

# Renormalization to localization without a small parameter

**Anton G. Kutlin⋆, and Ivan M. Khaymovich**

Max Planck Institute for the Physics of Complex Systems, D-01187 Dresden, Germany

⋆ anton.kutlin@gmail.com

## Abstract

We study the wave function localization properties in a $d$-dimensional model of randomly spaced particles with isotropic hopping potential depending solely on Euclidean interparticle distances. Due to generality of this model usually called Euclidean random matrix model, it arises naturally in various physical contexts such as studies of vibrational modes, artificial atomic systems, liquids and glasses, ultracold gases and photon localization phenomena. We generalize the known Burin-Levitov renormalization group approach, formulate universal conditions sufficient for localization in such models and inspect a striking equivalence of the wave function spatial decay between Euclidean random matrices and translation-invariant long-range lattice models with a diagonal disorder.

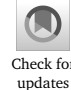

# 1   Introduction

After more than six decades of successful and intense study of Anderson localization (AL) passed from Anderson's seminal work [1] this field still embodies many puzzles and unexpected surprises such as the correspondence between many-body localized (MBL) systems and AL models on hierarchical structures like a random regular graph (RRG), including the presence in both counterparts of a whole phase of the subdiffusive wavepacket spreading in the finite range of parameters [2–6], which is absent in single-particle models on finite-dimensional lattices, and hot debates about the presence of a putative extended phase violating ergodicity [7–16] with non-trivial multifractal wavefunctions, claimed in other papers to be just a finite-size effect [17–27] due to a critical regime [13] close to the localization transition. Another surprise [1] of recent years in AL community is the presence of robust localization of all bulk eigenstates in long-ranged (e.g., dipolar) systems [32] beyond the convergence of the standard locator expansion and of the resonance counting [1, 33, 34]. For lattice models with diagonal disorder this phenomenon is directly linked in the literature to the effects of cooperative shielding [35] [2] and the emergence of localization due to correlations in hopping [36, 37] of these long-range models. However, this question for the models with off-diagonal disorder (e.g., due to disordered positions of lattice sites) is still open.

In this work, we address exactly this question via the discussion of the localization properties of systems described by Euclidean random matrices (ERM) [38], i.e. the systems of particles randomly distributed in $d$-dimensional space with the hopping of single-particle excitations, which solely depends on the Euclidean interparticle distance. Due to quite general description, ERMs cover a considerable class of physical models and arise naturally in various systems, e.g., in the ones with non-crystalline structures like gases, liquids, amorphous materials, and glasses. Although such models sometimes arise in the systems with short-range interactions, such as elastic networks [39], jammed soft spheres [40] or magnetic vortex plasma [41], more commonly ERMs are used to describe the long-range models. Indeed, long-range ERMs are applied to the analysis of the systems of particles with Coulomb interactions in two-dimensional irregular confinement [42], disordered classical Heisenberg magnets with uniform antiferromagnetic interactions [43], systems with dipole-dipole interactions such as dipolar gases [44], systems of ultracold Rydberg atoms [45] and so on. Even the effects of photon localization in atomic gases [46] are described by a long-range ERM. Although the ERM model itself was introduced [38] back in 1999 and its spectral properties are studied quite deeply [47–49], the analysis of the wave function properties, including their spatial structure and localization, crucial for above mentioned applications to physical models is barely carried out and represented in the literature only by a couple of numerical works [32, 50] or in quite restricted particular cases [51]. The present paper is aimed to fill this gap providing a generic analytical approach.

The problem with the analysis of the eigenstate structure in ERMs is caused by the absence of a small parameter. Indeed, unlike the models with the diagonal disorder, there is no way to treat the ERMs without the diagonal potential with the locator expansion approximation even for the infinitesimally small hopping term, due to the ideal resonance of all bare diagonal levels. This fact can be understood on the example of low-dimensional models with translation-invariant polynomial hopping which show localization of all bulk wavefunctions for any finite disorder either in the diagonal potential [32, 35, 36, 52] or in the position of the lattice sites [32], whereas in the complete absence of the disorder these models are translation-

---

[1]There are many other surprises such as emergence of multifractality in long-range static [28–30] or short-range driven [31] models with quasiperiodic potentials but we focus on the one relevant for our consideration.

[2]In this case a top energy level keeps delocalized even at strong disorder due to its energy diverging with the system size and shields the rest levels from the hopping terms.

invariant and, hence, delocalized. To overcome the above mentioned principal difficulty we generalized the known renormalization group (RG) approach (developed by Levitov [33, 34] and extended by Burin and Maksimov (BM) in [53] and by Mirlin and Evers in [54]) to the case of absence of a small parameter and to generic smooth Euclidean hopping term. Consequently, we show that for all ERMs with quite smooth potential [3] the bulk spectral eigenstates show localization.

The main idea behind this is similar to the one developed in [36], where the presence of (the measure zero of) delocalized states with energies diverging with the system size (not only the top energy level) at the either spectral edge gives the main contribution to the hopping term and is shown not to bring the system to the delocalization. In that case the effective hopping for the bulk spectral states can be obtained by the matrix-inversion trick developed in [36] which rewrites the eigenproblem in a special form, non-linear in eigenvalues, inverting all the high-energy contributions to the hopping term.

In the case of the current work on ERMs, the absence of a small parameter does not allow us to use the same technique and we have to develop a renormalization group (RG) approach. The resulting renormalization of the hopping terms is shown to evolve in such a way that the most of their spectral weight goes to the spectral edge states with energies increasing with the "renormalization scale" (system size) as in the matrix-inversion trick or the cooperative shielding. Thus, this significantly reduces the spectral weight of the hopping term in the bulk of the spectrum and localizes bulk spectral states.

## 2 Renormalization group approach

### 2.1 Main idea

The cornerstone idea of the renormalization group approach with respect to AL in random matrix problems [33, 34, 53, 54] is to rewrite the Hamiltonian of the system in such a form which is invariant under the iterative diagonalization procedure. The latter diagonalization procedure represents an elementary step of the renormalization scheme and thus should be done analytically as precise as possible. This often implies an exact diagonalization of certain $2 \times 2$ matrix blocks, which take into account most resonant levels hybridizing with the current one. This diagonalization procedure is crucially based on the assumption of isolated single resonant pairs of levels, which has a certain range of validity. The approximation can be formulated as follows: typically, for each iteration $i$ and any energy level represented by a diagonal matrix element $\varepsilon_n^i$ there is the only resonant level $\varepsilon_m^i$, $m \neq n$, such that the absolute value of the off-diagonal hopping element $t_{nm}^i$ between them is comparable or larger than the interlevel spacing $|\varepsilon_n^i - \varepsilon_m^i|$. The RG procedure diagonalizing the initial problem is formed by a set of consecutive elementary diagonalizations of the resonant level pairs.

Further we consider a random matrix Hamiltonian of a general form

$$H = \sum_i \varepsilon_i |c_i\rangle\langle c_i| + \sum_{i \neq j} f(r_{ij}) |c_i\rangle\langle c_j|, \tag{1}$$

with the deterministic real-valued function $f(r)$ which depends only on the Euclidean distance $r_{ij} = |r_{ij}| = |r_i - r_j|$ between some sites in $d$ dimensions, indices $i$ and $j$ numerate all $N$ $d$-dimensional sites $r_i$. The randomness in the model is given both by the off-diagonal elements through the positions of sites $r_i$ uniformly distributed in $d$-dimensional cube with the mean density equal to unity, and by the random bare on-site energies $\varepsilon_i$ with zero mean, dependent or independent of $f(r)$ and $r_i$. In particular, $\varepsilon_i$ could be even all equal to zero, as in the

---

[3]More rigorous general sufficient conditions are provided in the next sections.

power-law Euclidean (PLE) model considered in [32] with $d = 1$ and $f(r) = r^{-a}$. Unlike the models with translation-invariant hopping and only diagonal disorder [35–37, 53], the above model does not necessarily have a small parameter, and, hence, the approximation of the single resonances does not necessarily applicable from the first steps of RG. In terms of the wave functions it means that the localized ones (if any) can have extended "heads" of finite size $R_0$ which will be determined later, where eigenstate do not decay, but may, e.g., oscillate. These heads have a complex internal structure which cannot be obtained within RG approach, because at such distances the approximation of single resonances may fail. To overcome this difficulty, we introduce the following preliminary step before employing RG: we rewrite the Hamiltonian in a form $H = H_0 + V$ in such a way that, being expressed in the eigenbasis $|\varepsilon_n^0\rangle$ of $H_0$, it is invariant under the iterative diagonalization of resonant blocks and the approximation of single resonances is satisfied. This allows further RG treatment in the form of [33, 53, 54] and, thus, show the localization of the bulk of the states written in the basis $|\varepsilon_n^0\rangle$. If, in addition, eigenvectors of $H_0$ are exponentially localized in the initial basis $|c_i\rangle$, one can equally consider the localization and the eigenstate spatial structure in either of bases $|\varepsilon_n^0\rangle$ or $|c_i\rangle$. In this case one can forget about initial Hamiltonian and use the effective one instead.

To obtain the effective renormalizable Hamiltonian which satisfies all above mentioned conditions, we consider $H_0$ as the initial $H$ cut at $r_{ij} \leq R_0$

$$H_0 = \sum_i \varepsilon_i |c_i\rangle\langle c_i| + \sum_{r_{ij} \leq R_0} f(r_{ij})|c_i\rangle\langle c_j| \equiv \sum_n \varepsilon_n^0 |\varepsilon_n^0\rangle\langle \varepsilon_n^0| \tag{2}$$

and rewrite the original Hamiltonian in a form

$$H = \sum_n \varepsilon_n^0 |\varepsilon_n^0\rangle\langle \varepsilon_n^0| + \sum_{n,m} t_{nm}^0 |\varepsilon_n^0\rangle\langle \varepsilon_m^0|, \tag{3}$$

where $R_0$ is a cutoff radius at this zeroth step, and

$$t_{nm}^0 = \sum_{r_{ij} > R_0} f(r_{ij})\langle \varepsilon_n^0 |c_i\rangle\langle c_j|\varepsilon_m^0\rangle. \tag{4}$$

Since $H_0$ we used to obtain this form has short-range hopping in the original basis, the states $|\varepsilon_n^0\rangle$ are assumed to be localized with the localization scale of the order of $R_0$. Here we should note that even the worst case of the initial bare energies being strictly zero $\varepsilon_i = 0$, corresponding to all bare sites being in perfect resonance already at the first RG step, is covered by this method. Indeed, taking $R_0 = 1$ one can easily diagonalize $H_0$ and get (i) exponentially localized eigenstates $|\varepsilon_n^0\rangle$ and (ii) non-singular density of states (DOS) formed by nearly uncorrelated eigenvalues $\varepsilon_n^0$.

Further we restrict our consideration to the most relevant case of smoothly varying hopping potentials $f(r)$ at the scale $R_0$ [4] and neglect the difference between $r_{nm}$ and $r_{ij}$ due to localized nature of the wavefunctions $\langle \varepsilon_n^0 |c_i\rangle$ and $\langle c_j|\varepsilon_m^0\rangle$ at the cutoff radius $R_0 \ll r_{ij}, r_{nm}$, see Fig. 1. Within this approximation $\sum_j f(r_{ij})\psi_m^0(j) \simeq f(r_{im})\sum_j \psi_m^0(j)$, and the effective Hamiltonian, Eq. (3), takes the form

$$H_{\text{eff}} = \sum_n \varepsilon_n^0 |\varepsilon_n^0\rangle\langle \varepsilon_n^0| + \sum_{r_{nm} > R_0} q_n^0 q_m^0 f(r_{nm})|\varepsilon_n^0\rangle\langle \varepsilon_m^0|. \tag{5}$$

Here $q_n^0 = \sum_i \psi_n^0(i)$ is an effective "charge" of the state $|\varepsilon_n^0\rangle$, with $\psi_n^0(i) = \langle c_i|\varepsilon_n^0\rangle$. As we show below, this Hamiltonian is renormalizable, with the effective charges being the renormalization parameters. The zeroth-step cutoff radius $R_0$ should be determined in such a way that the approximation of the single resonances is valid for the first step of the renormalization group.

---

[4]See below for more rigorous conditions.

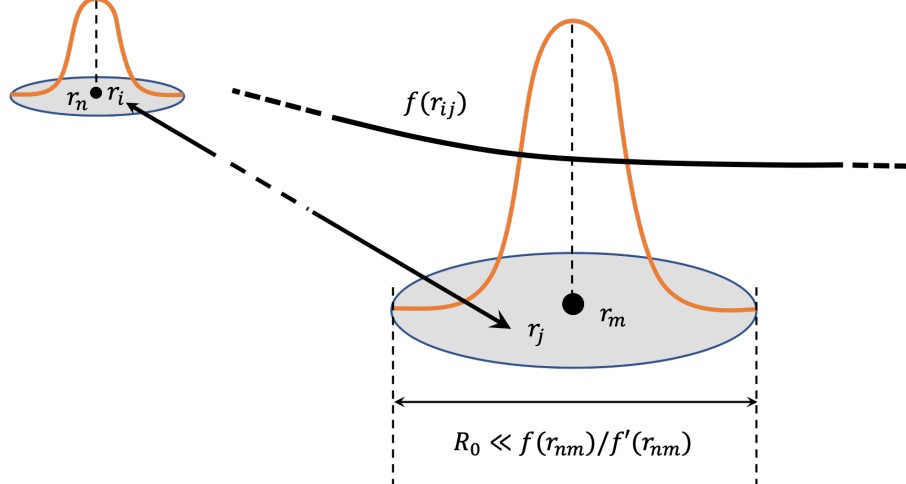

Figure 1: **Origin of the effective charge approximation.** Due to the smoothness of $f(r)$ and the localized nature of $\psi_m^0(j)$ one can approximately rewrite $\sum_j f(r_{ij})\psi_m^0(j)$ as $f(r_{im})q_m^0 = f(r_{im})\sum_j \psi_m^0(j)$.

First, we proceed to the renormalization group scheme which, from this point, is quite straightforward and leave the problem of the zeroth-step cutoff radius determination and the range of validity of the effective charge approximation for a further discussion. Assuming that on the $i$th iteration the renormalization group Hamiltonian $H_i$ has a form

$$H_i = \sum_n \varepsilon_n^i |\varepsilon_n^i\rangle\langle\varepsilon_n^i| + \sum_{R_i < r_{nm} \leq R_{i+1}} q_n^i q_m^i f(r_{nm})|\varepsilon_n^i\rangle\langle\varepsilon_m^i|, \tag{6}$$

with $R_{i+1} \gg R_i$ and the approximation of single resonances is valid, see Fig. 2, the next-step Hamiltonian $H_{i+1}$ can be written in the same form with renormalized eigenvalues $\varepsilon_n^{i+1}$ and charges $q_n^{i+1}$. Indeed, for each bare level both $\varepsilon_n^{i+1}$ and $q_n^{i+1}$ are (i) either equal to $\varepsilon_n^i$ and $q_n^i$ if at the current step there are no levels resonant with it or (ii) are determined by the

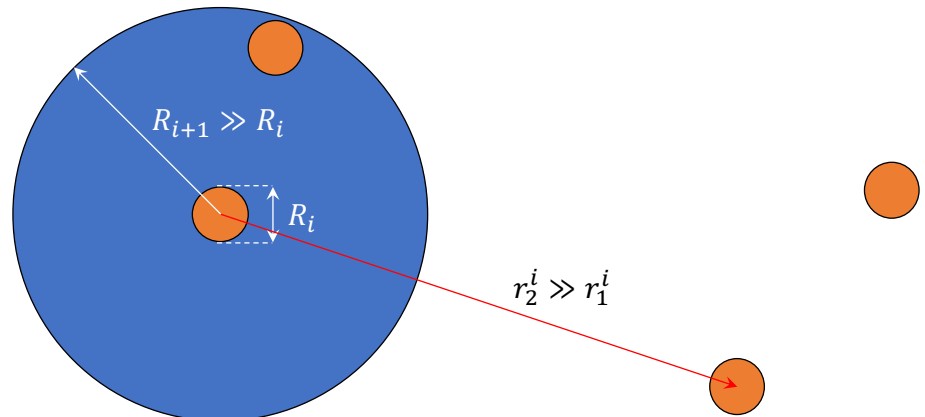

Figure 2: **Sketch of the single-resonance approximation.** For each eigenstate localized at $i$th step of RG at the radius $R_i$ (orange circles) the next cutoff radius $R_{i+1} = r_1^i$ (blue circle) is determined as the distance to the closest resonant level. Single-resonant approximation assumes that all other resonant levels are located much farther $r_2^i \gg r_1^i$.

diagonalization of the corresponding $2 \times 2$ resonant block coupling $\varepsilon_n^i$ and $\varepsilon_m^i$ levels

$$\varepsilon_{\pm}^{i+1} = \frac{1}{2}\left(\varepsilon_n^i + \varepsilon_m^i \pm \frac{\varepsilon_n^i - \varepsilon_m^i}{\cos\theta}\right), \tag{7a}$$

$$q_+^{i+1} = \cos\frac{\theta}{2}q_n^i + \sin\frac{\theta}{2}q_m^i, \quad q_-^{i+1} = -\sin\frac{\theta}{2}q_n^i + \cos\frac{\theta}{2}q_m^i, \tag{7b}$$

$$\tan\theta = 2\frac{q_n^i q_m^i f(r_{nm})}{\varepsilon_n^i - \varepsilon_m^i}, \quad -\pi/2 \le \theta \le \pi/2. \tag{7c}$$

This forms an elementary step of RG procedure which gives both the spectrum of the effective Hamiltonian $H_{\text{eff}}$, Eq. (5), and the asymptotic form of tails of its localized eigenstates [5]. Indeed, for $R_i \gg R_0$, when all the wavefunction heads are eventually formed, the strong resonances are rare and typical values of $\theta$ in (7c) are small. As a consequence, the typical wave functions transform as $|\varepsilon_{\pm}\rangle \simeq |\varepsilon_n\rangle \pm \theta/2|\varepsilon_m\rangle$, with $\langle c_j|\varepsilon_m\rangle$ being localized at $r_{jm} \simeq R_i$, see the right column of Fig. 3. Since $\theta \sim |q_n^i|^2 f(R_i)$ (a typical energy difference $\varepsilon_n^i - \varepsilon_m^i$ does not scale with $R_i$, see Appendix A), the tails are determined by the *effective hopping*

$$t_{\text{eff}}^i(\varepsilon) = \langle |q_\varepsilon^i|^2 \rangle f(R_i), \tag{8}$$

where

$$\langle |q_\varepsilon^i|^2 \rangle = \frac{\langle |q_n^i|^2 \delta(\varepsilon - \varepsilon_n^i)\rangle}{\nu_{R_i}(\varepsilon)} \tag{9}$$

is the squared effective charge for the state with energy $\varepsilon$ and averaging (denoted by $\langle\ldots\rangle$)

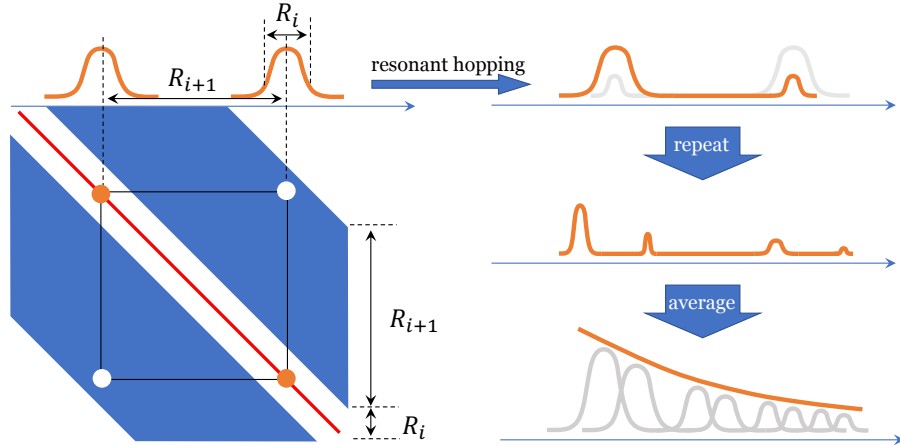

Figure 3: **Formation of the wavefunction tails by RG.** At $i$th step of RG bare eigenstates localized at distance $R_i$ (shown by orange curves in the top left and as orange circles on the diagonal of the matrix in the bottom left) are affected by hybridization via resonant hopping (white circles within blue non-resonant ones in the matrix) with eigenstates located at distance $R_{i+1}$. At later RG steps, $R_i \gg R_0$, the wavefunction hybridization is dominated by small angles $\theta$, Eq. (7c), determining the amplitude of the hybridized eigenstate at the distance $R_{i+1}$ with respect to the one localized at $R_i$ via the effective hopping, Eq. (8) (top right). Further steps of RG (middle right) and the disorder averaging (bottom right) form the typical wavefunction tails $\psi_n(j) = \langle c_j|\varepsilon_n\rangle \sim t_{\text{eff}}(r_{jn})$, Eq. (10).

---

[5]Like the ones in a numerical work [32] which have been found to be symmetric with respect to the critical value $a = d$ for $f(r) = r^{-a}$. In that paper it has been called the duality of the wave function power-law decay rate.

is taken over disorder realizations and index $n$, and $\nu_{R_i}(\varepsilon) = \langle \delta(\varepsilon - \varepsilon_n^i) \rangle$ is the density of states (DOS) at $i$th RG step. Note that the energy dependence of the effective hopping is not accidental as there are few delocalized states at the spectral edge for which the RG approach is not applicable.

In determination of the spatial decay we should take into account the difference in wavefunction averaging. For the typical averaging $\psi_{n,\text{typ}}^2(r_m) = \exp\left[\langle \ln \psi_n^2(r_m)\rangle\right]$ the eigenstate decays proportionally to $t_{\text{eff}}^i(\varepsilon)$, (8)

$$\psi_{\text{typ}}^2(R_i) \sim \left[t_{\text{eff}}^i(\varepsilon)\right]^2, \tag{10}$$

while for the mean averaging one have to take into account strong resonant contributions and obtain (due to Breit-Wigner-like profile of wavefunctions)

$$\langle \psi^2(R_i) \rangle \sim t_{\text{eff}}^i(\varepsilon). \tag{11}$$

## 2.2 Basic equations

To determine the evolution of the effective charges we first write the equation for the probability $P(q, \varepsilon; R)\mathrm{d}q\mathrm{d}\varepsilon$ of a state at a certain RG step with the cutoff radius $R$ to have energy and charge in the intervals $(\varepsilon, \varepsilon + \mathrm{d}\varepsilon)$ and $(q, q + \mathrm{d}q)$, respectively. Due to the hybridization (7) of resonant pairs the evolution of the probability distribution at one RG step takes the form

$$
\begin{aligned}
P(q, \varepsilon; R_2) - P(q, \varepsilon; R_1) = \frac{1}{2} \int & \mathrm{d}q_n \mathrm{d}\varepsilon_n P(q_n, \varepsilon_n; R_1) \mathrm{d}q_m \mathrm{d}\varepsilon_m P(q_m, \varepsilon_m; R_1) \\
\int_{R_1}^{R_2} \mathrm{d}^d \boldsymbol{r}_{nm} \Big( & \delta(\varepsilon - \varepsilon_+)\delta(q - q_+) + \delta(\varepsilon - \varepsilon_-)\delta(q - q_-) \\
& - \delta(\varepsilon - \varepsilon_n)\delta(q - q_n) - \delta(\varepsilon - \varepsilon_m)\delta(q - q_m) \Big).
\end{aligned} \tag{12}
$$

Here, for brevity, we omit upper indices $i$ and $i + 1$ and, instead of $R_i$ and $R_{i+1}$, write $R_1$ and $R_2$. The integration by $\mathrm{d}^d \boldsymbol{r}_{nm}$ is carried out over the whole region of the $d$-dimensional space in the interval $R_1 < r_{nm} < R_2$.

Equation (12) provides an exact recipe to calculate the distribution function $P(q, \varepsilon; R)$ at all steps of the renormalization scheme provided the approximations of effective charges and single resonances are valid. Needless to say that due to this exactness and an overall complex structure of the equation, its analytical solution is extremely tough to obtain without further approximations. However, since the quantities of primary interest are few first moments of the distribution function and not the distribution function itself, we can, using Eq. (12), write similar equations for the moments and then try to solve them, exactly or approximately. For example, it can be directly seen from Eq. (12) that the average eigenenergy $\langle \varepsilon \rangle$ and the magnitude of the average state charge $\langle q^2 \rangle$ do not change with the RG iterations at all

$$\langle \varepsilon \rangle = \int \mathrm{d}q \mathrm{d}\varepsilon P(q, \varepsilon; R)\varepsilon = \int \mathrm{d}\varepsilon \, \nu_R(\varepsilon)\varepsilon = \text{const}, \tag{13}$$

$$\langle q^2 \rangle = \int \mathrm{d}q \mathrm{d}\varepsilon P(q, \varepsilon; R)q^2 = \int \mathrm{d}\varepsilon \, \nu_R(\varepsilon)\langle |q_\varepsilon(R)|^2 \rangle = \text{const}. \tag{14}$$

Note that the latter equality is exact (even beyond RG consideration) and equal to the unity $\langle q^2 \rangle = 1$ due to the completeness of the eigenbasis at each $i$th RG step and for every single realization. Due to Eq. (13) the value of $\langle \varepsilon \rangle$ is completely determined by the zeroth-step cutoff Hamiltonian $H_0$ or, in other words, by the heads of the wavefunctions.

## 2.3 Equation for effective charges

In order to determine the effective hopping, one can write the equation for $\chi_\varepsilon(R) = \langle|q_\varepsilon(R)|^2\rangle \nu_R(\varepsilon) = \int dq P(q, \varepsilon; R) q^2$ [6] which is an energy-dependent second $q$-moment of $P(q, \varepsilon; R)$, straightforwardly following from Eq. (12)

$$
\chi_\varepsilon(R_2) - \chi_\varepsilon(R_1) = \frac{1}{2} \int dq_n d\varepsilon_n dq_m d\varepsilon_m P(q_n, \varepsilon_n; R_1) P(q_m, \varepsilon_m; R_1)
$$

$$
\int_{R_1}^{R_2} d^d r_{nm} \left( \delta(\varepsilon - \varepsilon_+) q_+^2 + \delta(\varepsilon - \varepsilon_-) q_-^2 - \delta(\varepsilon - \varepsilon_n) q_n^2 - \delta(\varepsilon - \varepsilon_m) q_m^2 \right).
$$

(15)

Clearly, the last two delta-functions give after integration $-\chi_\varepsilon(R_1) C_d (R_2^d - R_1^d)$ where $C_d = \pi^{d/2}/\Gamma(1 + d/2)$ is a volume of $d$-dimensional ball of a radius 1, $\Gamma(x)$ is the Gamma-function, so we concentrate on the contributions $J_\pm$ from the first two ones corresponding to $\delta(\varepsilon - \varepsilon_\pm)$. After changing of integration variables from $\varepsilon_n$ and $\varepsilon_m$ to $w = (\varepsilon_n + \varepsilon_m)/2$ in both integrals and $t_\pm = \pm q_n q_m f(r) \cot(\theta/2)$ in $J_\pm$, respectively, one can integrate out the remaining delta-functions and simplify integrands to the identical expressions for $J_+$ and $J_-$,

$$
J_\pm = \frac{1}{2} \int dq_n dq_m \int_{R_1}^{R_2} d^d r \int_{|t| > |q_n q_m f(r)|} dt \left( q_m^2 + \frac{2q_n^2 q_m^2 f(r)}{t} + \frac{q_n^4 q_m^2 f^2(r)}{t^2} \right) \times
$$

$$
P(q_n, \varepsilon - t; R_1) P\left( q_m, \varepsilon - \frac{q_n^2 q_m^2 f^2(r)}{t}; R_1 \right).
$$

(16)

The fact that the integration excludes small values of $t$ allows us to simplify the exact relation in the approximation of the small charges. Indeed, assuming an existence of $R$-dependent cutoff $Q(R)$ for $q$ such that the distribution function $P(q, \varepsilon; R)$ is exponentially small for $q > Q(R)$ and $Q^2(R) f(R)$ is small compared to a width of DOS, $\nu_R(\varepsilon)$, for $R > R_0$, one can neglect small terms both in the argument of the second $P(q, \varepsilon; R)$ and in the first brackets getting

$$
J_\pm \simeq \frac{1}{2} \int dq_n dq_m \int_{R_1}^{R_2} d^d r \fint dt \left( q_m^2 + \frac{2q_n^2 q_m^2 f(r)}{t} \right) P(q_n, \varepsilon - t; R_1) P(q_m, \varepsilon; R_1).
$$

(17)

Here $\fint$ denotes the principle value integration from $-\infty$ to $\infty$. The function $Q^2(R)$ in the approximation formulation can be replaced by $\langle|q_\varepsilon(R)|^2\rangle$ to obtain a sufficient condition for the validity of the approximation. Thus, the sufficient condition for the approximation to be valid is to have effective hopping $t_{\text{eff}}(R) = \langle|q_\varepsilon(R)|^2\rangle f(R)$ decaying to zero at infinite distances. As we show below, $t_{\text{eff}}(R)$ always behave so in the range of validity of RG scheme, i.e. provided the approximations of effective charges and single resonances are valid.

Assuming r.h.s. of Eq. (15) to be sufficiently small even for significantly different $R_1$ and $R_2$, one can replace the finite-difference equation for $\chi_\varepsilon$ by the following differential one as by taking formally the limit $R_2 \to R_1$

$$
\frac{\partial \chi_\varepsilon(\xi)}{\partial \xi} = 2\chi_\varepsilon(\xi) \fint \frac{dz \chi_z(\xi)}{\varepsilon - z}, \quad \xi = \int_{R_0}^{R} f(r) d^d r.
$$

(18)

Here, $\xi$ is a natural renormalization scale variable. Solving this equation, one obtains

$$
\chi_\varepsilon(\xi) = \frac{\chi_0(\varepsilon)}{(1 - k_0(\varepsilon)\xi)^2},
$$

(19)

---

[6]Since the density of states doesn't depend on cutoff radius, see Appendix A

where $\chi_0(\varepsilon) = \chi_\varepsilon(\xi = 0) = \chi_\varepsilon(R = R_0)$, and $k_0(\varepsilon)$ is determined by the expression

$$k_0(\varepsilon) = \oint \frac{\mathrm{d}z\,\chi_0(z)}{\varepsilon - z}. \tag{20}$$

The condition for replacing the finite-difference equation (15) by the differential one (18) can be rewritten as

$$k_0(\varepsilon)\chi_0(\varepsilon) \ll 1. \tag{21}$$

This is the only validity criterion which explicitly differentiate states by their energy [7].The condition will later lead to the mobility edge estimation.

As seen from (19), the renormalization of $\chi_\varepsilon(\xi)$ and, thus, of the hopping $t_{\mathrm{eff}}^i$, Eq. (8), is *non-trivial* only if $\xi(R)$ goes to infinity as $R \to \infty$. Otherwise, the effective hopping is proportional to the original one, $t_\varepsilon^{\mathrm{eff}}(R) \propto f(R)$. Instead, in the case of non-trivial renormalization $\chi_\varepsilon(\xi) \sim \chi_0(\varepsilon)/k_0(\varepsilon)^2\xi^2$ for $k_0\xi \gg 1$, and

$$|\psi_{\mathrm{typ}}(R)| \sim \langle \psi^2(R) \rangle \sim t_\varepsilon^{\mathrm{eff}}(R) \sim \frac{\chi_0(\varepsilon)f(R)}{\nu_R(\varepsilon)k_0(\varepsilon)^2\xi^2(R)} \propto \frac{f(R)}{\left(\int^R f(r)\mathrm{d}^d\boldsymbol{r}\right)^2}, \tag{22}$$

which shows that our original model (2) is dual to the other model, with $\tilde{f}(R) \propto t_\varepsilon^{\mathrm{eff}}(R)$ instead of $f(R)$ and localized eigenstates. This result can be applied to any smooth function $f(r)$ and, thus, claims the localization of the bulk spectral states for all long-range ERMs with smooth potential. For example, in a particular case of $f(r) = r^{-a}$, our result explains the duality of the wave function decay

$$|\psi_{\mathrm{typ}}(R)| \sim R^{-\mu(a)}, \ \mu(a) = \mu(2d - a), \tag{23}$$

with respect to the critical point $a = d$ observed for $d = 1$ in [32]. The latter model will be discussed in details in Sec. 3.

Note that from Eq. (19) one can easily see that the latter approximation might break down at certain small positive $k_0(\varepsilon)$ due to the presence of a pole which is incompatible with the requirement (21). Thus, at each cutoff $R$ there is a certain energy $\varepsilon^*$ determined by the vicinity of the pole $k_0(\varepsilon^*) = 1/\xi(R)$ at which $\chi_\varepsilon(\xi)$ can take unbounded values. It signals that the approximation of single resonances with a given $R_0$ breaks down for these energies and the states may become delocalized. On the other hand, we know that, due to the general relation $\langle q^2 \rangle = \int \mathrm{d}\varepsilon\,\chi_\varepsilon(R) = 1$ working at any $R$ including $R_0$ the function $\chi_0(\varepsilon)$ should be integrable to unity. As a result, $\chi_0(\varepsilon)$ should have a sharp peak at the energies $\varepsilon^*(R_0)$ in the following interval

$$\varepsilon_{min}^*(R_0) \leq \varepsilon^*(R_0) \leq \varepsilon_{max}^*(R_0), \tag{24a}$$

$$\varepsilon_{min}^*(R_0) \sim \langle \varepsilon^0 \rangle, \tag{24b}$$

$$\varepsilon_{max}^*(R_0) \sim \int^{R_0} f(r)\mathrm{d}^d\boldsymbol{r}. \tag{24c}$$

Indeed, since $\chi_0(\varepsilon) = \nu_0(\varepsilon)\langle |q_\varepsilon(R_0)|^2 \rangle$, its maximum lies between the absolute maxima of the density of states and the squared effective charge function. The lower bound $\varepsilon_{min}^*(R_0)$ is of the order of the mean energy which doesn't scale with $R_0$, while the upper bound $\varepsilon_{max}^*(R_0)$

---

[7] Due to the single resonance approximation which forms the very basis of Eq. (12), it is applicable only if its r.h.s is small, i.e. when the probability density function $P(q, \varepsilon, R)$ changes slowly with the renormalization scale. So, the condition (21) is deeper than just the mathematical trick to go from differences to differentials.

can be estimated as the energy of the trial state with $R_0$ components all equal to $R_0^{-d/2}$ with the same sign (zero-momentum plane wave), eventually leading to (24c). This state is chosen as an estimate because it gives the maximal possible value of $\langle |q(R_0)|^2 \rangle$ for the normalized state with $R_0$ non-zero components

$$\langle |q(R_0)|^2 \rangle_{max} \sim R_0^d. \tag{25}$$

Moreover, it is natural to assume that such a state is close to the eigenstate of the model for spatially homogeneous distribution of sites as the fluctuations site positions are averaged out on this zero-momentum plane wave.

It is important to note that the presence in such a system of the states with the large effective charge (maybe not only the above mentioned one) which leads to their delocalization causes the localization of the bulk spectral states (similar to the matrix-inversion trick [36]) as the main spectral weight of $\chi_\varepsilon(R)$ is absorbed by these high-energy states. As we will show in the last section in some models (like in PLE) these delocalized states may be located not at the very spectral edge and this severely questions an alternative cooperative shielding explanation present in the literature [35, 52].

Now we are in the position when we are ready to check our approximations, find the range of validity of our method and estimate the size of the wavefunction head $R_0$.

## 2.4 Effective-charge approximation

We start with the conditions on the hopping function $f(r)$ required for the effective charge approximation (5). The initial hopping term (4) between states $|\varepsilon_n^0\rangle$ and $|\varepsilon_m^0\rangle$ reads as

$$t_{nm}^0 = \sum_{r_{ij} > R_0} f(r_{ij}) \langle \varepsilon_n^0 | c_i \rangle \langle c_j | \varepsilon_m^0 \rangle. \tag{26}$$

To go from this form to the approximate one with effective charges we have to assume that $f(r_{ij})$ differs only slightly from $f(r_{nm})$ for any such $i$ and $j$ that $\psi_n^0(i)$ and $\psi_m^0(j)$ have significantly non-zero values, i.e. for $r_{ni}, r_{mj} < R_0$, Fig. 1. Mathematically, for an isotropic model it gives the following condition:

$$R_0 \frac{df(r)}{dr}\bigg|_{r=r_{nm}} \ll f(r_{nm}). \tag{27}$$

Since the only hopping terms which matter for the RG approach are the ones from the resonant blocks, the distance $r_{nm}$ in the condition should be of the order of a typical distance between counted single resonances. As shown in the next subsection, the next cutoff $R_{i+1}$ should be chosen to be much smaller than the average distance between single resonances in the full (not truncated) Hamiltonian and, hence, [8]the validity of the effective charge approximation is governed by the condition [9]

$$R_i \frac{df(r)}{dr}\bigg|_{r=R_{i+1}} \ll f(R_{i+1}). \tag{28}$$

---

[8] The requirement for the function $f(r)$ to be smooth in points corresponding to the typical distances between resonances, Eq. (28), limits it to have all its sufficiently strong singularities to be located at small distances determined by the zero-cutoff radius, $r < R_0$. By the term 'sufficiently strong' we mean such singularities that alter the typical distance between resonances moving it from the value $R_{i+1}$ towards the vicinity of the pole. Indeed, as soon as the typical resonance is caused by the singular hopping values rather than by $R_{i+1}$ the corresponding hopping terms cannot be approximated by the effective charges and the whole RG approach fails.

[9] The presented condition is sufficient, but far from necessary. An actual necessary and sufficient condition has to deal with meaning of relative fluctuations of $f(r)$ in the $R_i$−vicinity of the typical distance between resonances on the $i$th step. This fact actually allows the same RG treatment not only for the ERM models with smooth deterministic $f(r)$, but also for the models with hopping terms of the form $t_{ij} = (1 + h_{ij})f(r_{ij})$ with $h_{ij}$ being a random variable with zero mean and relatively narrow distribution function, $f^2(r_{ij})\langle h_{ij}^2 \rangle \ll r^{-2d}$ (similar to [37]).

From this relation it is clear that it puts the restrictions not only on the function $f(r)$ but also determines how $R_i$ and $R_{i+1}$ have to be related for a given $f(r)$. For example, in the case of PLE, $f(r) \sim r^{-a}$, Eq. (28) gives $R_{i+1} \gg R_i$. If this restriction contradicts to any other one, then the whole RG approach fails and it may bring the bulk eigenstates of the system to the delocalization.

## 2.5 Single-resonance approximation

Next, we consider the range of validity of the single-resonance approximation for the effective Hamiltonian (5). To justify the approximation, we count the resonances on the $i$th RG step and estimate the probability for the multiple resonances to occur. For $i$th RG step, a number of states $N_\varepsilon^i(r)$ separated by a distance $r$, $R_i < r < R$, from a certain state with energy $\varepsilon$ and resonant to it can be written as

$$N_\varepsilon^i(R) = \int_{R_i}^{R} p_\varepsilon^i(r) \rho(\boldsymbol{r}) \mathrm{d}^d \boldsymbol{r}, \tag{29}$$

via

$$p_\varepsilon^i(r) = \int_{-\langle |q_\varepsilon^i|^2 \rangle f(r)}^{\langle |q_\varepsilon^i|^2 \rangle f(r)} \mathrm{d}\varepsilon' \, \nu_i(\varepsilon + \varepsilon'), \tag{30}$$

the probability to have a single resonance at distance $r$ and via the density of states $\nu_i(\varepsilon)$ with energies $\varepsilon_n^i$ at a given $i$th RG step. Here $\rho(\boldsymbol{r})$ is the average density of sites $\boldsymbol{r}$ in the spatial region of integration. For simplicity we consider the models with uniform spatial density. Thus, we rescale it to unity, $\rho(\boldsymbol{r}) \equiv 1$, and omit in further expressions.

Probabilities (30) help us to define the typical probability [10]to have no resonances in the layer $R_i < r < R$,

$$P_{0,\varepsilon}^i(R) = \exp\left( \int_{R_i}^{R} \ln(1 - p_\varepsilon^i(r)) \mathrm{d}^d \boldsymbol{r} \right), \tag{31}$$

and the typical probability to have exactly one resonance in that layer

$$P_{1,\varepsilon}^i(R) = P_{0,\varepsilon}^i(R) \int_{R_i}^{R} \frac{p_\varepsilon^i(r)}{1 - p_\varepsilon^i(r)} \mathrm{d}^d \boldsymbol{r}. \tag{32}$$

For the single-resonance approximation to be valid, $R_{i+1}$ has to be chosen in such a way that the probability to have more than one resonance in the layer, $R_i < r < R_{i+1}$, is small compared to the probability to have exactly one resonance, i.e.

$$1 - P_{0,\varepsilon}^i(R_{i+1}) - P_{1,\varepsilon}^i(R_{i+1}) \ll P_{1,\varepsilon}^i(R_{i+1}). \tag{33}$$

Assuming the probability $p_\varepsilon^i(r)$ to be small compared with unity in all points of the layer one can approximately write

$$P_{0,\varepsilon}^i(R) \sim \mathrm{e}^{-N_\varepsilon^i(R_{i+1})}, \quad P_{1,\varepsilon}^i(R) \sim N_\varepsilon^i(R_{i+1}) \mathrm{e}^{-N_\varepsilon^i(R_{i+1})}, \tag{34}$$

which finally gives us the smallness, $N_\varepsilon^i(R_{i+1}) \ll 1$, of the number of resonant states in the layer, Eq. (29), as a requirement. The latter requirement can be written solely via renormalization

---

[10] Expressions Eqs. (31) and (32) are valid for quite large layers, $R_i < r < R$, with sufficiently small fluctuations of spatial density of sites. An actual probability to have no resonances, of course, is equal to $\prod_k (1 - p_\varepsilon^i(r_{jk}))$ and depends on the particular realization of disorder as well as on the particular choice of $\boldsymbol{r}_k$.

scales $\xi(R_i)$ and $\xi(R_{i+1})$ in the case of the non-singular $R$-independent DOS (see Appendix A confirming this assumption) and the decreasing function $f(r)$

$$2\chi_\varepsilon(R_i)(\xi(R_{i+1}) - \xi(R_i)) \ll 1 \quad \Rightarrow \quad \xi(R_{i+1}) \ll \xi^2(R_i). \tag{35}$$

Here we used the definition $\chi_\varepsilon(R) = \langle |q_\varepsilon(R)|^2 \rangle \nu_R(\varepsilon)$, the asymptotic expression for the probability of single resonances $p_\varepsilon^i(r) \simeq 2\chi_\varepsilon(R_i)f(r)$ for $R_i, R_{i+1} \gg 1$, and the expression (19). Equation (35) together with the condition (21) and Eq. (28) form a complete set of requirements for the RG to be applicable.

The breakdown of the single-resonance approximation occurs when the typical spectrum width is comparable with the effective hopping strength. Indeed, in that case, due to the normalization of the density of states, $p_\varepsilon^i(R_i) \sim 1$ (see Eq. (30)) and, consequently, there is no such $R_{i+1} > R_i$ to satisfy the condition $N_\varepsilon^i(R_{i+1}) \ll 1$. This result can be intuitively understood as follows: to have a negligible probability of multi-resonance collision we should have low probability of the two-resonance collision as well. So, from this point of view, the first step of our renormalization procedure and an introduction of the zero-cutoff radius were made to remove the singularity of $\nu_R(\varepsilon)$ and made it shallow enough on the scale of the effective hopping.

In the next section we test RG scheme and the above approximations together with wavefunction localization properties for the particular case of the PLE model.

## 3   Power-law Euclidean model

To show the validity of our approximations we apply the approach developed in the previous section to the one-dimensional power-law Euclidean (PLE) model and compare our analytical results with numerics. The model is defined by a Hamiltonian (1) with $\varepsilon_i = 0$

$$H_{PLE} = \sum_{i \neq j} \frac{|c_i\rangle\langle c_j|}{|\boldsymbol{r}_i - \boldsymbol{r}_j|^a}. \tag{36}$$

As it follows from the previous numerical studies of this model [32], its wave functions show a striking duality, $a \to 2d - a$, of the bulk wave function decay rate. These eigenstates are polynomially localized $\psi_{n,typ}(r_m)^2 \sim |r_n - r_m|^{-2\mu}$ with the exponent $\mu = \max\{a, 2d - a\}$ for all positive values of the parameter $a$. Although this model is very similar to the one of Burin and Maksimov (BM) [53] considered also in [36, 52, 55], the above wavefunction duality in PLE model has not yet been explained theoretically as the matrix inversion trick invented by one of us in [36] breaks down in the absence of diagonal disorder.

The RG approach developed in the present paper provides the desired explanation. Indeed, for $f(r) \sim r^{-a}$, the non-trivial renormalization occurs for $a < d$, Eq. (18), giving $t_\varepsilon^{\text{eff}}(r) \propto r^{a-2d}$ according to (8), while for $a > d$ renormalization scale $\xi$ converges with $R$ and the standard perturbative approach works giving the polynomial decay of the form $r^{-a}$. The results for the spatial decay of typical, Eq. (10), and mean, Eq. (11), wave function tails for several cutoff values $R$ corresponding to RG procedure are shown in Fig. 4.

Consider this model in more details. First of all, Eq. (35) in case of PLE model give $R_{i+1} \ll R_i^2$, which is compatible with the approximation of effective charges provided $R_i \gg 1$ as $R_i^2 \gg R_i$ should be valid. It means that the zero-cutoff radius $R_0$ is finite and large compared to the unity. Fig. 4 provides the following estimate of the cutoff radius $R_0 \sim 30$, which is in full agreement with the above consideration. However, our RG approach is actually applicable only for $a > 0$: for negative values of $a$ the approximation of single resonances breaks down since the original hopping $f(r)$ is no longer a decreasing function. Nevertheless, according

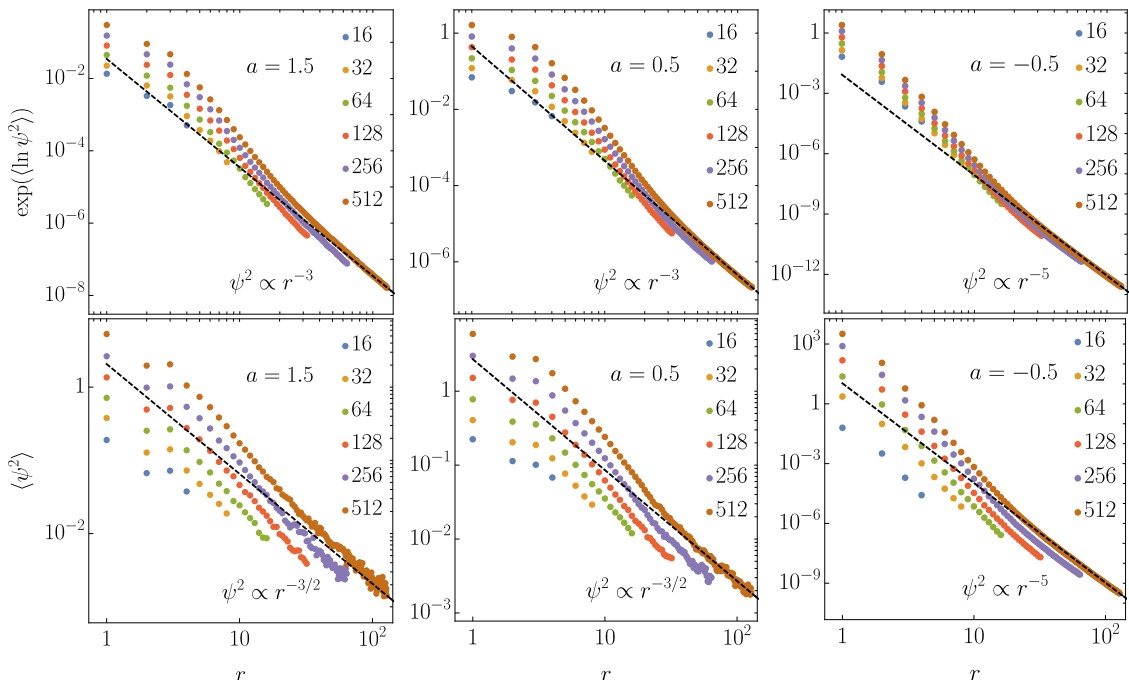

Figure 4: **The spatial decay of mid-spectrum eigenstates in PLE model.** (Upper row) typical $\ln \psi^2_{\text{typ}}(r_{nm}) = \langle \ln \psi^2_n(r_m) \rangle$ and (Lower row) mean $\langle \psi^2_n(r_m) \rangle$ wavefunction power-law decay for several powers $a$ (shown in labels) and cutoffs $R$ (shown in legend). All points are averaged over $10^3$ disorder realizations and shifted vertically for clarity. Dashed lines show analytical predictions, Eqs. (10) and (11) (written in panels as equations). The right column shows that the validity of RG scheme for typical wavefunction decay can be extended also to some spatially increasing (though unphysical) hopping.

to numerical results, this breakdown of the RG approach doesn't lead to the delocalization or even to the aforementioned duality breaking for typical wavefunction spatial decay. This fact may be caused by the destructive interference of the resonances or, in other words, by the higher-order corrections to the perturbation series.

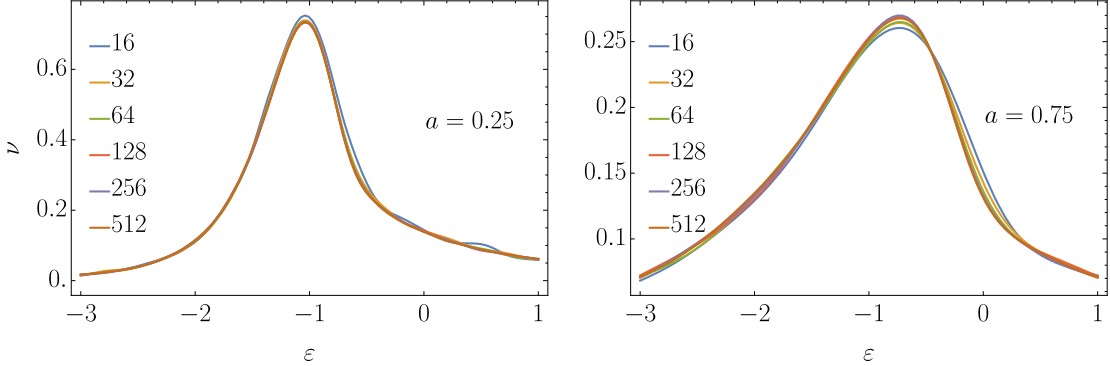

Figure 5: **Density of states for several cutoffs averaged over $10^3$ realizations.** All plots show that DOS saturates at very small cuttoff values $R$ (shown in the legend) and it is a non-singular function.

After determining the validity range of RG, we check numerically the fact about the density of states stated in the Appendix A. According to the RG approach, the function $\nu_R(\varepsilon)$ barely

depends on the cutoff radius $R$, $a < d$:

$$\frac{d\nu_R(\varepsilon)}{dR} \propto R^{a-2d} \tag{37}$$

and it converges to a non-singular function. Both these statements are clearly seen from Fig. 5 supporting the analysis done in the Appendix A.

Next, although the estimates and numerical results presented above in this section justify the RG approach for PLE model in general, they do not provide any information about the approximations we made going from Eq. (15) to Eq. (18), i.e., the small-charge approximation and the approximation of the finite-difference equation by the differential one. To check that the effective charges are indeed behave according to Eq. (19), we calculate it explicitly for a set of different cutoff values, see Fig. 6. The lower row of panels shows the function $\langle|q_\varepsilon(R)|^2\rangle\xi^2(R)$ which does not depend on the cutoff value and collapses to a universal curve with good accuracy. Moreover, the insets show that the above collapse works relatively well until the maximum of $\langle|q_\varepsilon(R)|^2\rangle$, but not only in the bulk of the spectrum.

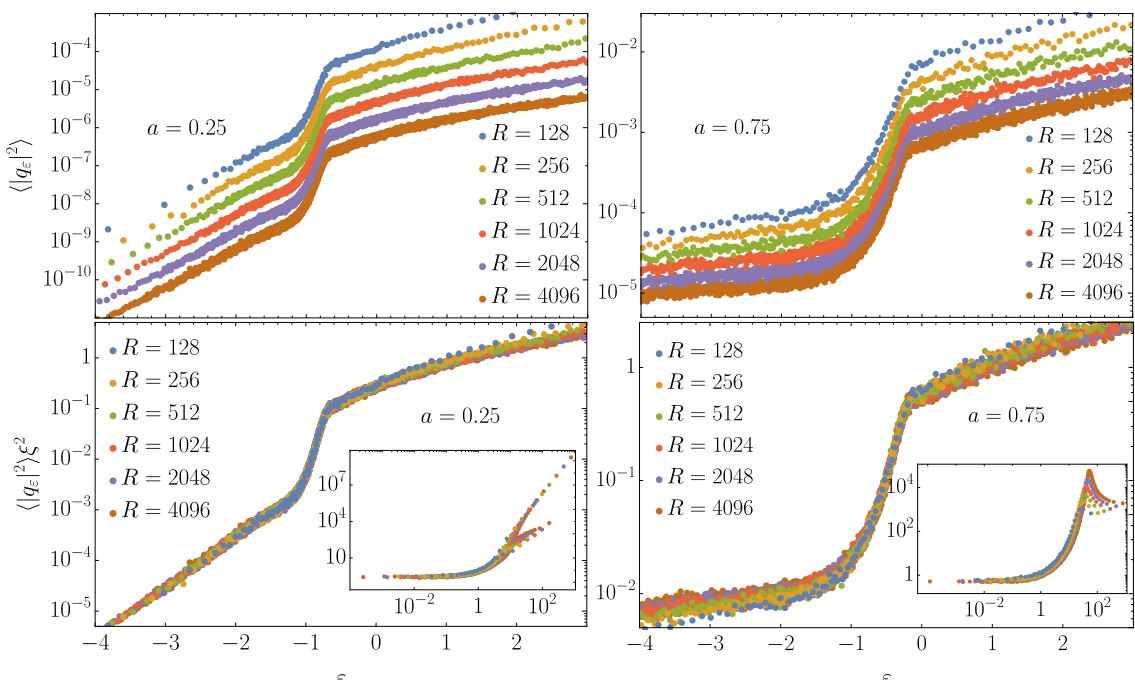

Figure 6: **Mean squared effective charge** $\langle|q_\varepsilon(R)|^2\rangle$ **versus energy** $\varepsilon$. (Upper row) energy dependence of $\langle|q_\varepsilon(R)|^2\rangle$ for several cutoffs (shown in legend) in the vicinity of the DOS maximum. (Lower row) energy dependence of $\langle|q_\varepsilon(R)|^2\rangle\xi^2(R)$ collapsed by the multiplication by the squared renormalization scale, confirming the analytical result, Eq. (19). (Insets) the same collapse for all positive energies in log-log scale. In all panels the data is averaged over $10^3$ realizations.

Finally, our theory predicts that, for $a < d$, $\langle|q_\varepsilon(R)|^2\rangle$ as a function of $\varepsilon$ must have a sharp maximum at $\varepsilon_{max}^* \sim R_0^{d-a}$ with the magnitude of the order of $R_0^d$, see (24) and the corresponding discussion. By combining these two estimates we get the one describing the energy dependence of the maximal magnitude with increasing cutoff $R$

$$\langle|q_\varepsilon(R)|^2\rangle_{max} \propto \varepsilon^{\frac{d}{d-a}}. \tag{38}$$

As shown in the insets to Fig. 7, this is exactly the case, at least for $d = 1$.

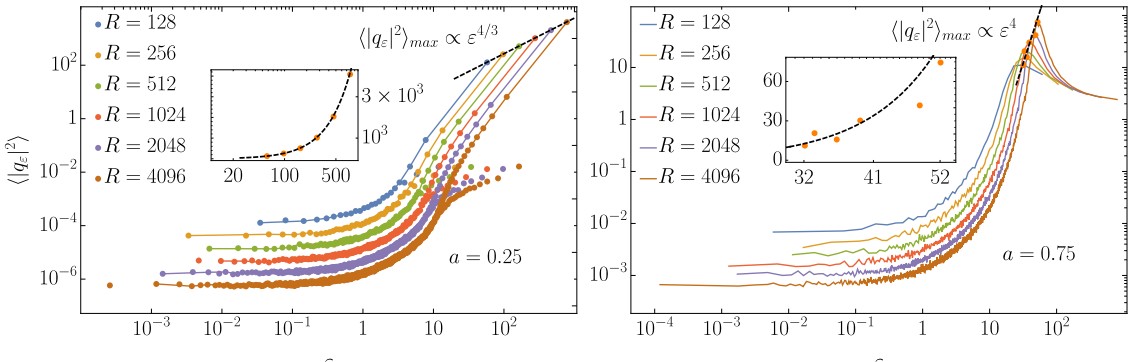

Figure 7: **Energy dependence of the maximal effective charge.** The dots show the same $\langle|q_\varepsilon(R)|^2\rangle$ for all positive energies in log-log scale as in the insets to Fig. 6, but without collapsing, black dashed lines show the evolution of the maximal $\langle|q_\varepsilon(R)|^2\rangle$ and its energy with the increasing cutoff radius according to Eqs. (24c), (25) and (38). The insets show the same maximal points of $\langle|q_\varepsilon(R)|^2\rangle$ in linear scale with the power-law black dashed fitting curves coinciding with the ones in the main panels. The data is averaged over $10^3$ realizations.

One may notice that the maximum of the effective charge $\langle|q_\varepsilon|^2\rangle$ in Fig. 7 occurs at the edge of the spectrum for $a < d/(d+1) = 1/2$, while for $a > 1/2$ the maximal energy corresponds to the constant $R$-independent value of $\langle|q_\varepsilon|^2\rangle = O(1)$ [11]. The explanation of this is based on the fact that on top of the trial delocalized state there are rare states localized at few adjacent sites. For $R_0^d$ sites the minimal distance between such sites, typical for each disorder realization, is given by $r_{\min} = R_0^{-d}$ [12] and thus the energy of the state localized on this pair of states scales as $\varepsilon \sim f(r_{\min}) \sim R_0^{da}$ and at $a > d/(d+1)$ these states will form the edge of the spectrum. The corresponding effective charges for these states are given by the expression $\lim_{\varepsilon\to\infty}\langle|q_\varepsilon|^2\rangle = 2$.

As mentioned in the first section, the presence of the delocalized states with large energies scaling with the system size (or cutoff value) causes the localization in long-range Euclidean matrices in the similar way as in the models with diagonal disorder and translation-invariant hopping terms [36] due to the leakage of most of the charge spectral weight to large energies (and measure zero of states). The lesson which one should take from this is the following: it is *not* the ground (or anti-ground) state with the energy diverging with the system size which matters for the localization of the bulk spectrum (like in the cooperative shielding approach [35, 52]), but the presence of high-energy delocalized states (with high effective charges) do this job. The latter high energy states do not need to be at the very spectral edge, see the right panel of Fig. 7.

---

[11] Note that the case $a < d/(d+1) = 1/2$ characterized by the delocalized eigenstates at the very spectral edge is an artefact of finite statistics in our numerical simulations. Indeed, in the renormalization group written for the infinite system with a certain cutoff, see, e.g., the bottom left of Fig. 3, has to be determined by the infinite number of states localized at few very close sites, which, in turn, form the very spectral edge. In numerics instead of the cutoff $R_i$ of the infinite matrix we diagonalize full $R_i \times R_i$ matrices removing many two-site localized states. Another effect of such numerics is that it forces the localization radii to be not larger than $R_i$ at each $i$th RG step and reduces the corresponding finite-size effects for the wavefunction tails (shown in Fig. 1).

[12] This estimate is given by solution of the equation $R_0^d P(r_{\min}) = 1$, with the distribution of distances between adjacent sites, homogeneously distributed in $d$-dimensional space with unit density, given by Poisson formula, $P(r) \sim re^{-r}$.

# 4 Conclusion and discussions

To sum up, in this work we develop a generic renormalization group (RG) approach applicable to a wide range of Euclidean random matrix models, which shows localization of the bulk mid-spectrum states and provides the wave function decay for these states in an explicit form, Eq. (8). The range of validity of the above statements is governed by three conditions: the applicability of effective charge approximation (28) restricting the hopping potential $f(r)$ to be smooth, the single-resonance approximation (35) which is satisfied, e.g., for the bounded monotonically decaying function $f(r)$, and the slow probability density evolution condition (21) which is deeply interconnected with the single resonances approximation.

The above mentioned requirements allow us to get rid of *any* small parameter, which is crucial for the standard RG approach [33, 34, 53, 54], and show the renormalization to the localization for all spectral bulk eigenstates. This localization is solely caused by the drastic spectral flow of the renormalized effective hopping to high-energy delocalized states (forming measure zero of all states in low dimensions $d \leq 2$). The developed RG has many similarities to the so-called matrix inversion trick developed by one of us in [36] and complements and extends it to the case of Euclidean matrices with off-diagonal disorder (with or even without diagonal disordered part).

Moreover, the developed approach shows the equivalence between Euclidean models and translation invariant models with diagonal disorder and smooth hopping potential $f(r)$ not only in the localization properties, but also in the spatial decay of bulk mid-spectrum eigenstates. We believe that this equivalence can be generalized to non-smooth and even anisotropic hopping which is recently under the spotlight [52, 55], but this is the topic of further investigation.

# Acknowledgements

We thank A. L. Burin for helpful and stimulating discussions.

**Funding information** I. M. K. acknowledges the support of German Research Foundation (DFG) Grant No. KH 425/1-1 and the Russian Foundation for Basic Research Grant No. 17-52-12044.

# A The RG evolution of the density of states

To make sure that the shape of the density of states $\nu_R(\varepsilon)$ barely depends on the renormalization scale, we write its renormalization equation in the similar way as for the effective charges (15)

$$
\begin{aligned}
\nu_{R_2}(\varepsilon) - \nu_{R_1}(\varepsilon) = \Bigg( & \int dq_n dq_m \int_{R_1}^{R_2} d^d\boldsymbol{r} \int_{|t|>|q_n q_m f(r)|} dt \left( 1 + \frac{q_n^2 q_m^2 f^2(r)}{t^2} \right) \\
& P(q_n, \varepsilon_n; R_1) P\left( q_m, \varepsilon_m - \frac{q_n^2 q_m^2 f^2(r)}{t}; R_1 \right) - \nu_{R_1}(\varepsilon) C_d (R_2^d - R_1^d) \Bigg).
\end{aligned}
\tag{39}
$$

The point is that the r.h.s. of the latter equation vanishes in the first-order approximation in small charges giving the first indication of the fact how weak the renormalization scale dependence is. To proceed further, one needs to make tricky assumptions about an analytical

structure of the distribution function $P(q, \varepsilon; R)$ and expand it up to at least the first order of the Taylor series assuming $q_n^2 q_m^2 f^2(r)/t$ to be small. As a result, we get the differential equation

$$\frac{\partial \nu_R(\varepsilon)}{\partial R} = -\frac{R^{d-1} f^2(R)}{4\chi_\varepsilon(R)} \frac{\partial \chi_e(R)}{\partial \varepsilon} \frac{\partial \chi_e(R)}{\partial \xi(R)}. \tag{40}$$

After substituting here the approximate expression for $\chi_\varepsilon$ from Eq. (19), we find that, in case of large $\xi(R)$ for large $R$, the derivative $\partial_R \nu_R(\varepsilon) \sim R^{d-1} f^2(R)/\xi^3(R)$, i.e.,

$$\frac{\partial \nu_R(\varepsilon)}{\partial R} \propto f(R)\frac{\mathrm{d}}{\mathrm{d}R}\xi^{-2}(R). \tag{41}$$

As long as $f(R)$ goes to zero with increasing $R$, the r.h.s of the latter expression does the same. This concludes that, for $R \gg R_0$, the function $\nu_R(\varepsilon)$ is saturated and only slightly differs from its limiting value $\nu(\varepsilon) = \nu_\infty(\varepsilon)$.

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
