# Peer review of "Renormalization to localization without a small parameter"

_SciPost Physics, doi:SciPost Phys. 8, 049 (2020)_

## Round 1 · Referee Report · Alexander Burin (Referee 1) · 2020-2-4

Strengths

This is very interesting work resolving the problem of Anderson localization in the presence of the long-range hopping of constant sign. The authors has performed a combination of analytical and numerical studies and characterized localized and delocalized states. Numerics agrees with analytics so the results can be extrapolated to numrically inaccessible regimes

Weaknesses

Fig. 7 is confusing. I cannot find any demonstration of the predicted dependence of the maximum charge on energy. I think the graph needs major improvement to make it clear for readers.
I would also suggest to add more numerical characterization of ejgen-states including inverse participation ratio and level statistics to give the reader more knowledge about the system behavior.

Report

As I noticed in my opinion this is very interesting work that deserves publication after the weaknesses will be addressed.

Requested changes

I would recommend to improve Fig. 7 and add graphs for inverse participation ratio and level statistics.

---

## Round 1 · Referee Report · Anonymous (Referee 2) · 2020-2-18

Strengths

I recommend the publication of this very interesting article.

Weaknesses

The text refers to many previous works but it is sometimes too allusive.
Two Examples :
a) the 'matrix-inversion trick [19]' is mentioned several times (p3, p10,p16) but is never explained.
b) before Eq 22 : "the generalization of the duality [15,19]" deserves some explanation about what precise duality is meant here.

So I feel that the paper would be much more interesting for a broader audience if it were more self-contained and could be read without having to look at too many previous references. In addition, the authors should state more clearly what is similar and what is different from the previous works.

Report

Here are some comments that the authors might consider in order to improve their manuscript :

1) The authors use "renormalization time" (p3) , " natural renormalization time variable" (before Eq 19), etc… while it is related to the size R : I think that the authors should avoid the word 'time' (since nowadays there are actually many works on the dynamics in localized systems) and use some more accurate name like 'RG scale' or equivalent.

2) I think that the arguments given in the numerous long notes [37-38-39-40],[42-43] should be included in the main text instead of being buried among citations.

Requested changes

English : the last sentence of the Introduction (Part 1) starting with " this significantly reduces …" is not clear and should be rephrased.

---

## Round 2 · Author Response

Dear Editor,

Thank you for providing us the reports both of Prof. Burin and of the second Referee.
We would like to resubmit the article entitled “Renormalization to localization without a small parameter” for consideration in SciPost Physics.

First of all, we would like to thank both Referees for their careful reading of the manuscript, for providing constructive criticism, and for their positive comments. We hope that our responses and changes made to the manuscript have convinced both Prof. Burin and the second Referee so that the manuscript is now considered suitable for the publication in SciPost Physics.

Yours sincerely,

Anton G. Kutlin and Ivan M. Khaymovich

---

## Round 2 · List of Changes

In order to improve the manuscript, we follow the recommendations of the Referees and make the following changes in the manuscript:
1. Following the comment of Prof. Burin, we have modified Fig. 7 by showing the points of the maxima of all curves in the linear scale in the insets. We emphasize the same points in the main panels by full circles and the same dashed fitting lines as in the insets.
2. Following the first comment of the second Referee, we have reduced the number of repetitive citations of the previous papers and explained the terms “matrix-inversion trick” and “the duality”.
3. We have also clarified in the introduction what is similar and what is different in our paper from the previous works.
4. The “renormalization time” has been replaced by the “renormalization scale” throughout the text.
5. The footnotes have been moved from the Reference list to the text.
6. The phrase starting with "this significantly reduces …" has been corrected and the corresponding discussion has been added.
7. In addition, we have added some relevant references and discussed the surprises of the Anderson and many-body localization in a little bit more detailed manner.
Regarding the suggestion of Prof. Burin to add more numerical data for the PLE model, we added a reference to the paper [32] which already contains all the data. We think that inverse participation ratio and level statistics do not directly connect to the discussed analytical method and will unnecessarily complicate and enlarge the paper.

You are currently on this page

Resubmission 2001.06493v2 on 20 March 2020

---

## Editorial Decision

published